# Decreased Risk of Knee Osteoarthritis with Taller Height in an East Asian Population: A Nationwide Cohort Study

**DOI:** 10.3390/jcm13010092

**Published:** 2023-12-23

**Authors:** Dong Hwan Lee, Hwa Sung Lee, Soo Hyun Jang, Jun-Young Heu, Kyungdo Han, Se-Won Lee

**Affiliations:** 1Department of Orthopedic Surgery, Yeouido St. Mary’s Hospital, College of Medicine, The Catholic University of Korea, 10, 63-Ro, Seoul 07345, Republic of Korea; ldh850606@naver.com (D.H.L.); eagleddong@naver.com (S.H.J.); 2Department of Orthopedic Surgery, Incheon St. Mary’s Hospital, College of Medicine, The Catholic University of Korea, 56, Dongsu-ro, Incheon 21431, Republic of Korea; med@live.co.kr; 3Department of Statistics and Actuarial Science, Soongsil University, 369, Sangdo-ro, Seoul 06978, Republic of Korea

**Keywords:** height, taller height, osteoarthritis, risk of osteoarthritis, nationwide study

## Abstract

Background: Numerous studies have explored factors impacting osteoarthritis (OA), but its relationship with height remains uncertain. This study investigates the relationship between height and osteoarthritis risk in South Korea. Methods: Participants aged 50 or older who underwent health screenings in 2009 were selected from the National Health Insurance System database. A total of 1,138,904 subjects were divided into height quartiles (Q1–Q4) based on age and gender. Cox proportional hazard models were used to assess knee osteoarthritis incidence risk, with the shortest quartile (Q1) as the reference. Results: After adjusting for age, sex, income, smoking, drinking, exercise, hypertension, diabetes mellitus, dyslipidemia, and body mass index (BMI), no significant difference in OA incidence risk based on height was observed. However, when adjusted for weight instead of BMI, we observed a gradual decrease in hazard ratio with increasing height. The hazard ratio for the tallest group was 0.787 (95% CI, 0.781~0.795). Similar results were obtained in all subgroups. Conclusions: Compared to previous studies, our findings present a clear distinction. Therefore, there may be racial differences in the association between height and knee OA risk, and our study provides evidence that, in East Asian populations, taller individuals have a reduced risk of knee OA.

## 1. Introduction

Osteoarthritis (OA) has become a growing concern in regions with rapidly aging societies such as East Asia, where it is the most common joint disorder [1,2,3]. In the past, OA was thought to be primarily a condition caused by repetitive mechanical loading that leads to the wear of articular cartilage. However, recent research has focused on the relevance of chronic low-grade inflammation [4,5], and an increasing emphasis on genetic studies has improved the understanding of OA pathophysiology [6,7,8]. The occurrence and progression of OA are now known to be influenced by various factors, and associations with other diseases are also being uncovered due to a plethora of research in diverse fields. Among these studies, big data analysis based on accumulated data has been actively pursued. Research on weight, body mass index (BMI), and obesity as basic physiological indicators has been extensively published [9,10,11,12], as well as studies on exercise-related factors such as physical activity and muscle mass [13,14]. There have been numerous studies on the association of OA with other diseases, such as diabetes mellitus (DM). Various big data analyses related to OA have been presented, addressing the frequency of OA occurrence in individuals with DM, as well as studies on treatment and prognosis [15,16]. Additionally, studies have explored associations with cardiovascular disease [17], dementia [18], mental health [19], and even polycystic ovary syndrome [20]. However, studies on the association between height and OA through nationwide big data analysis are limited, with only a few studies incorporating height among various risk factors [21,22,23].

Unlike weight, obesity, and BMI, which are intuitively suspected to affect OA incidence, studies on the association between height and OA are rare. Some studies have reported an increase in the occurrence of OA with taller height, and the authors conducted research using cohorts from Finland and Norway [21,24]. On the other hand, other studies have found an increase in OA with shorter height, and this research was based on a dataset from Florida, a state with diverse racial demographics [25]. There also are studies reporting no association between height and knee OA, and one of them was conducted on patients in Hong Kong [26]. Such diverse results make it difficult to draw definitive conclusions on their association. Several genetic studies have reported that the GDF5 gene influences the association between height and OA [7,27,28]. One study identified specific alleles showing high differentiation in East Asian populations which were associated with an increased risk of OA and decreased height [6]. 

Based on the results of previous studies, including genetic research, we assumed ethnic differences in the association between height and knee OA risk. However, there has been a lack of big data analysis on the association between height and knee OA risk in the East Asian population. Therefore, in this study, we aimed to investigate the association between adult height and the risk of knee OA using a nationwide dataset from Republic of Korea.

## 2. Method

### 2.1. Data Source and Study Population

We used the Korean National Health Insurance System (NHIS) database, which is a social medical insurance system operated by the Korean government. The majority of the population is obligated to enroll in this medical insurance system, which requires individuals aged 40 and above to undergo health screenings every 1 or 2 years. We selected our study population and obtained data from the health screening database and the NHIS claims dataset. The NHIS claims dataset is organized based on the International Classification of Disease (ICD)-10 codes and provides information on examinations and treatments that can be used for population-based cohort studies [29]. We collected data on individuals aged 50 and above who underwent health screenings in 2009. Using the NHIS claims dataset, we identified cases of knee OA among the study population. The diagnosis of knee OA was based on a combination of diagnostic codes (M15, M17, M19) and the use of plain knee radiography by referencing validation studies [30]. We excluded patients who had a diagnosis of knee OA before 2009 or had any missing data. Additionally, patients who were diagnosed with knee OA within one year from the health screening date were also excluded. The exclusion of the latter is because it is challenging to establish an association when knee OA is diagnosed within one year of the screening. Additionally, this situation may lead to potential issues of reverse causation. Finally, a total of 1,138,904 individuals were included in this retrospective cohort study (Figure 1). This cohort was followed until the time when patients were diagnosed with knee OA in the NHIS claims dataset or until the last follow-up date, which was 31 December 2020.

### 2.2. Ethics

The protocol of this study was reviewed and approved by the ethics committee of our hospital and was conducted following the principles of the Declaration of Helsinki. To assess information from the NHI database, permission was obtained from the Korean NHIS. As the data were anonymized, there was no need for informed consent.

### 2.3. Data Collection and Comorbidities 

Health screenings consist of both survey responses and direct measurements from clinical tests. Information on health-related lifestyle habits, such as smoking, drinking, physical activity, and income level, was collected through surveys. Measurements of height, weight, and blood pressure were performed by healthcare professionals. Height was measured without shoes, and weight was measured with lightweight clothing. BMI was calculated by dividing weight (kg) by the square of height (m^2^). Blood samples were collected after an 8 h fasting period. 

To remove any compounding effects, the identified comorbidities, including hypertension (HTN), DM, and dyslipidemia, were defined as follows: HTN was defined as ICD-10 codes I10-13 or I15 and at least one anti-hypertensive drug prescription or as systolic/diastolic blood pressure ≥ 140/90 mmHg. DM was defined as ICD-10 codes E11-14 and at least one anti-diabetic drug prescription or fasting plasma glucose ≥ 126 mg/dL. Dyslipidemia was defined as ICD-10 code E78 and a lipid-lowering agent prescription or fasting total cholesterol ≥ 240 mg/dL.

### 2.4. Statistical Analysis

Baseline characteristics were presented as mean ± standard deviation for continuous variables or as number and percentage for categorical variables. Continuous variables were compared using analysis of variance (ANOVA), and categorical variables were compared using the chi-square test. The incidence rate of knee OA was calculated by dividing the number of events by the total person-years of follow-up. The participants were stratified into quartiles (Q1–Q4) based on height criteria according to age and sex. Age groups were divided into 50–59 years, 60–69 years, 70–79 years, and 80 years and above. As age increased, the average height decreased, resulting in lower height thresholds for the quartiles (Table 1). If the height difference between quartiles is set too large, there is an issue where the number of patients included in the tallest group becomes too small. Therefore, cut-off values were established as shown in Table 1. The height difference between the Q1 group and Q4 group was set to a minimum of 8cm, and considering the average height of patients, we deemed this to be a meaningful difference. The Cox proportional hazards model was used to assess the risk of knee OA in relation to height. Several models were used for adjusting variables: Model 1 (no adjustment), Model 2 (adjusted for age and sex), Model 3 (adjusted for age, sex, income, smoking, drinking, exercise, HTN, DM, and dyslipidemia), Model 4 (Model 3 + BMI), and Model 5 (Model 3 + weight). We used the Cox proportional hazards model to calculate the hazard ratio (HR) and 95% confidence interval (CI) for each height quartile, with the lowest quartile as the reference. Subgroup analyses were based on age and sex. We validated the proportional hazards assumption using the log–log cumulative survival plot. For all statistical analyses, we used SAS version 9.4 (SAS Institute, Cary, NC, USA), with *p* < 0.05 considered significant.

## 3. Results

### 3.1. Baseline Characteristics 

A total of 1,138,904 participants was divided into quartiles (Q1–Q4) based on height, and the baseline characteristics for each quartile are summarized in Table 2. Although there were differences in the total number of participants among the quartiles, most lifestyle indicators, comorbidities, smoking, and drinking percentages were similar. As height increased, weight also increased, but there was little difference in BMI among the height quartiles. 

### 3.2. Risk of Osteoarthritis According to Height

The incidence of OA was analyzed using Cox proportional hazards models for each height quartile (Q1–Q4). The HR was calculated based on the smallest quartile (Q1 group) as the reference. The results from the five models are presented in Table 3. In Models 1, 2, and 3, no significant difference or trend was observed among the Q1–Q4 groups. In Model 4, after adjusting for BMI, the HR and 95% CI for the Q2 group were 0.99 (0.982–0.998); for the Q3 group, they were 0.988 (0.979–0.996); and for the Q4 group, they were 0.996 (0.988–1.004), indicating no significant difference in the risk of OA incidence among the Q1–Q4 groups. However, in Model 5, which was adjusted for weight instead of BMI, the risk of OA decreased significantly as height increased. In Model 5, the HR and 95% CI for the Q2 group were 0.908 (0.9–0.915); for the Q3 group, they were 0.853 (0.846–0.86); and for the Q4 group, they were 0.787 (0.781–0.795). The Kaplan–Meier estimations for the probability of OA incidence by height quartile are presented in Figure 2.

### 3.3. Subgroup Analysis by Gender and Age 

Subgroup analyses were conducted based on gender. The results of Cox proportional hazard models for each gender and height quartile (Q1-Q4) are presented in Table 3. In the male group, for Model 4, the HR and 95% CI for the Q2 group were 0.971 (0.96–0.983); for the Q3 group, they were 0.963 (0.951–0.975); and for the Q4 group, they were 0.964 (0.952–0.975). In the male group, for Model 5, the HR and 95% CI for the Q2 group were 0.904 (0.893–0.915); for the Q3 group, they were 0.855 (0.844–0.866); and for the Q4 group, they were 0.796 (0.786–0.807). In the female group, for Model 4, the HR and 95% CI for the Q2 group were 1.013 (1.002–1.024); for the Q3 group, they were 1.005 (0.995–1.017); and for the Q4 group, they were 1.035 (1.023–1.046). In the female group, for Model 5, the HR and 95% CI for the Q2 group were 0.917 (0.907–0.927); for the Q3 group, they were 0.848 (0.839–0.858); and for the Q4 group, they were 0.787 (0.778–0.796). In Model 5 adjusted for BMI, for males, the risk of OA occurrence was slightly lower in the Q2–Q4 groups compared to the Q1 group. For females, the risk of OA occurrence was slightly higher in the Q2–Q4 groups compared to the Q1 group. In both genders, the differences were too small to be considered statistically significant. However, in Model 4, after adjusting for weight, both male and female participants showed a significant decrease in OA incidence risk with increasing height, with no significant difference between genders.

Subgroup analysis by age, divided into four categories (50–59 years, 60–69 years, 70–79 years, and 80 years and older), was conducted using the same approach (Table 4). The age groups were as follows: 671,733 individuals aged 50–59 years, 331,849 individuals aged 60–69 years, 118,864 individuals aged 70–79 years, and 16,458 individuals aged 80 years and above. In Model 4, after adjusting for BMI, the HR and 95% CI for Q4 were 0.983 (0.972–0.994) for the 50–59 age group, 0.991 (0.977–1.005) for the 60–69 age group, 1.049 (1.024–1.074) for the 70–79 age group, and 1.125 (1.041–1.216) for the 80 years and older group. In Model 5, after adjusting for weight, the HR and 95% CI for Q4 were 0.778 (0.769–0.787) for the 50–59 age group, 0.785 (0.774–0.797) for the 60–69 age group, 0.83 (0.81–0.851) for the 70–79 age group, and 0.891 (0.824–0.963) for the 80 years and older group. As with the previous results, after adjusting for BMI in Model 4, height was not significantly associated with the OA incidence risk in all age groups except the 80 years and older group. However, after adjusting for weight in Model 5, the risk of OA decreased with increasing height in all age groups, and the magnitude of the decrease diminished with age. In both Model 4 and Model 5, the risk of OA increased with age.

## 4. Discussion

Various studies have reported associations between height and various diseases, including conditions that may seem unrelated, such as atrial fibrillation, meningioma, and skin cancer [31,32,33,34]. However, there have been relatively few studies on the relationship between height and OA, and the results have been diverse, making it difficult to draw conclusive findings. Although it might be assumed intuitively that height would not have a significant impact on OA, after considering demonstrated associations with various other conditions, we conducted this study. Numerous studies have analyzed various risk factors for OA occurrence, and big data studies focusing on the risk of OA associated with weight and BMI have been published extensively. However, to the best of our knowledge, this study represents the first investigation using a nationwide dataset to explore the risk of knee OA based on height. And through this study, we confirmed that in the East Asian population, taller height is associated with a reduced risk of knee OA incidence. Additionally, although there were slight differences depending on gender and age, the overall trend showed a decrease in the risk of knee OA with increasing height.

Welling et al. [24] conducted a prospective cohort study on the Northern Finland Birth Cohort 1966 (NFBC) to analyze the risk of OA in the knee or hip according to height. Their study used statistical analysis methods similar to ours; for knee OA, 202 participants were ultimately included. The participants were divided into four groups based on height, and the risk of OA incidence was evaluated using a Cox proportional hazards regression model with the shortest height group as the reference. After adjusting for BMI, education, smoking, and activity, the tallest height group showed an HR for knee OA of 2.5 for males and 1.8 for females. Additionally, after adjusting for weight, education, smoking, and activity, the HR for knee OA in the tallest height group was 1.7 for males and 1.1 for females. When adjusting for BMI, OA risk significantly increased in both genders with height. However, when adjusting for weight, males showed a substantial increase in OA risk with taller height, whereas females exhibited only a slight increase. These results differed significantly from our study, in which we also categorized participants into quartiles based on height. We conducted analyses using Model 4, which adjusted for age, sex, income, smoking, drinking, exercise, HTN, DM, dyslipidemia, and BMI, and using Model 5, which adjusted for Model 3 plus weight instead of BMI. In Model 4, for males, the HR for the tallest Q4 group was 0.964; that in Model 5 was 0.796. For females, the HR in Model 4 for the Q4 group was 1.035, and that in Model 5 was 0.787. Unlike the study by Welling et al. [24], where OA risk significantly increased after adjusting for BMI, our research found no significant increase in risk, and, surprisingly, when adjusting for weight, OA risk showed a significant reduction compared to their study.

Moreover, in the study by Welling et al. [24], the risk of OA did not increase sequentially as height increased. However, in our study, after adjusting for weight, we observed a sequential decrease in the risk of OA occurrence as height increased. In a subgroup analysis of participants divided into four groups based on age, adjusting for weight yielded similar results; OA risk decreased with increasing height. Age showed a tendency to increase OA risk, which is an expected finding. In gender-based subgroup analysis, although our study found no significant difference in OA risk between males and females, a study by Welling et al. showed a substantial difference. However, these discrepancies might be attributed to the different cutoffs used to divide the participants based on height. 

We speculate that these contradictory results could be influenced by ethnic differences. As previously mentioned, several studies have shown the association of height with various diseases. Height is determined by a complex interplay of genetic and nutritional factors, and its association with various diseases is probably influenced significantly by genetic factors. Genetic studies on OA have also been actively conducted. In multiple studies, the GDF5 gene has been reported to play a significant role in the occurrence of OA [28,35,36,37]. In particular, the GDF5 rs143383 polymorphism has been found to be associated with knee OA [38,39]. The GDF5 gene has also been found to be associated with height, and in this context, the GDF5 rs143383 polymorphism appears to play a significant role as well [27,39]. Additionally, genetic expression and effects of such genes vary by ethnicity, with GDF5 polymorphisms having a stronger association with knee OA in East Asian populations compared to Caucasian populations [39]. Wu et al. recently reported that rs143383 and rs148834, which are related to decreased height and OA risk, showed positive selection in East Asian populations [6]. Due to the ethnicity-dependent variations in the GDF5 gene polymorphism, the association between the risk of OA and height may show contrasting results among racial groups. However, there is insufficient evidence to firmly establish these genetic factors as the definitive basis for the differences. Nevertheless, this study, conducted using a nationwide dataset from a single ethnic group, holds more statistical significance compared to previous research. Moreover, by adjusting for various confounding factors such as age, sex, income, smoking, drinking, exercise, HTN, DM, and dyslipidemia, the study can provide more meaningful and reliable results. Therefore, at the very least, it can be concluded that, in East Asian populations, the risk of OA decreases with an increase in height.

Our study has several limitations. First, reverse causation is a potential issue. However, we considered a washout period and excluded subjects who were diagnosed with knee OA within one year from their health screening date. By doing so, we minimized the potential bias associated with this issue. Second, since we used NHIS data, we lacked access to radiological data. Additionally, we were unable to access medical data, which means we could not include factors such as history of trauma or specific underlying medical conditions in our analysis. Third, due to difficulties in reflecting socioeconomic status, there might be an inadequate assessment of diet and labor factors. Individuals from lower socioeconomic backgrounds may experience poorer nutrition and shorter stature. It is also speculated that they are more likely to engage in physically demanding labor. In such cases, the risk of OA occurrence is likely to increase. Taking these factors into consideration could potentially make the study even more valuable. We adjusted for information on income from health examination surveys to minimize this limitation. Fourth, in this study, the differences in cut-off values between quartiles were not substantial. If we had set the height differences between groups to be greater than this study, there was a concern that the number of patients in the largest group would significantly decrease, potentially rendering the analysis meaningless. Therefore, we opted to set the cut-off values in this manner. However, the minimum 8cm height difference between the Q1 and Q4 groups is considered clinically significant. In the results, there was a reduction of over 20% in the risk of knee osteoarthritis (OA) in the Q4 group. Therefore, this is deemed sufficiently meaningful. Last, we attempted to compare our results with those of the NFBC study, but the difference in age groups between the two cohorts poses a challenge. However, since both studies calculated HR based on the group with the shortest height within a specific age range, we find it reasonable to compare them. Furthermore, OA is generally more likely to occur in middle-aged and older populations, making the age range selected for this study suitable for the analysis.

In conclusion, our study revealed that, in an East Asian population, a taller height is associated with a reduced risk of knee OA incidence. Although there were slight differences depending on gender and age, the overall trend of decreased OA risk with increasing height was consistent. The analysis was conducted by adjusting for age, sex, income, smoking, drinking, exercise, HTN, DM, dyslipidemia, and weight or BMI. When adjusting for weight, an increase in height was associated with a significant decrease in the risk of OA occurrence. However, when adjusting for BMI, there was no increase in the risk of OA. These results demonstrate a distinct difference from the NFBC study. Future studies, such as genetic analyses, should be conducted to clarify the reasons behind these disparities. Additionally, more comprehensive nationwide data studies across diverse ethnicities could shed light on potential racial variations in the influence of height on OA risk.

## Figures and Tables

**Figure 1 jcm-13-00092-f001:**
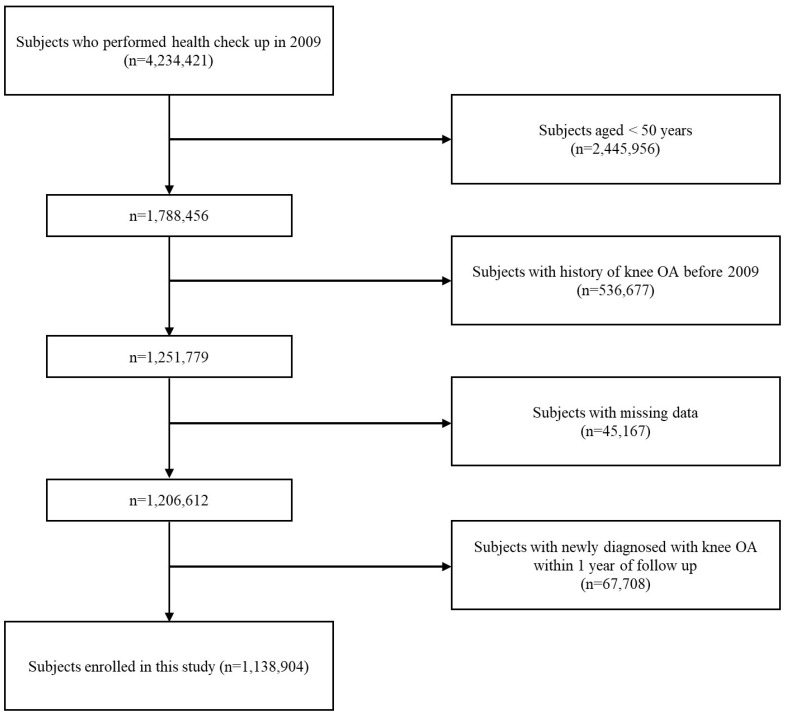
Flow chart of cohort selection.

**Figure 2 jcm-13-00092-f002:**
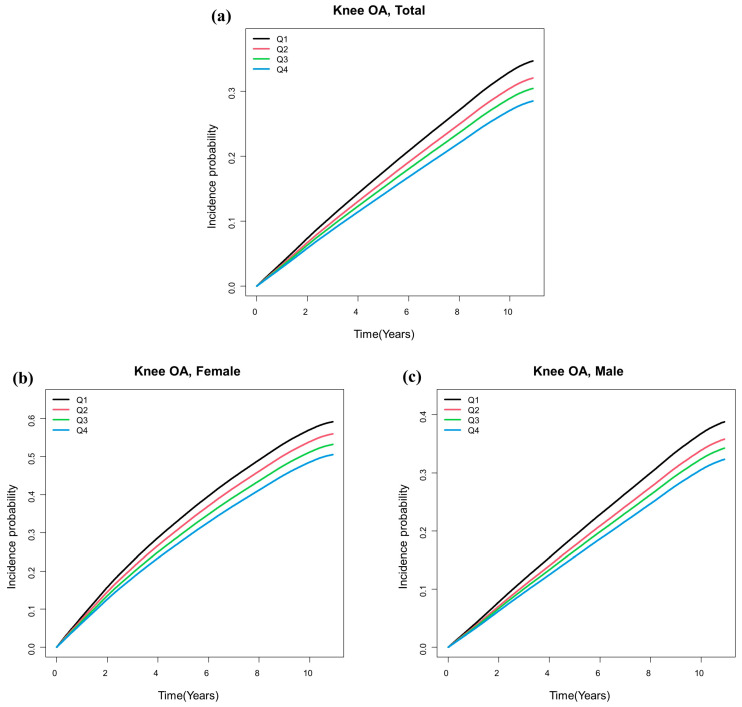
Kaplan–Meier estimates of cumulative incidence of osteoarthritis according to height (Q1–Q4). (**a**) Total, for both females and males. (**b**) For females. (**c**) For males. In all graphs (**a**–**c**), it can be observed that as height increases, the incidence of knee osteoarthritis decreases.

**Table 1 jcm-13-00092-t001:** Height quartile cut-off values by gender and age.

**Age (years)**	50–59	60–69	70–79	80–
**Male**	Height quartile cut-off values (cm)			
**Q1**	–164	–162	–160	–158
**Q2**	165–168	163–166	161–164	159–162
**Q3**	169–171	167–169	165–168	163–166
**Q4**	172–	170–	169–	167–
**Female**	Height quartile cut-off values (cm)			
**Q1**	–152	–149	–146	–142
**Q2**	153–155	150–153	147–150	143–146
**Q3**	156–159	154–156	151–153	147–150
**Q4**	160–	157–	154–	151–

**Table 2 jcm-13-00092-t002:** Baseline characteristics of subjects.

Male					
	Q1 (*n* = 172,404)	Q2 (*n* = 179,332)	Q3 (*n* = 137,611)	Q4 (*n* = 173,542)	*p* Value
**Age, years (%)**					<0.0001
50–59	98,749 (57.28)	102,159 (56.97)	75,291 (54.71)	99,367 (57.26)	
60–69	53,061 (30.78)	55,217 (30.79)	40,572 (29.48)	54,307 (31.29)	
70–79	18,335 (10.63)	19,597 (10.93)	19,490 (14.16)	17,535 (10.1)	
≥80	2259 (1.31)	2359 (1.32)	2258 (1.64)	2333 (1.34)	
**Income, low 25 (%)**	40,539 (23.51)	39,360 (21.95)	28,573 (20.76)	34,817 (20.06)	<0.0001
**DM (%)**	29,629 (17.19)	31,253 (17.43)	24,120 (17.53)	30,456 (17.55)	0.0211
**Hypertension (%)**	75,394 (43.73)	77,580 (43.26)	59,601 (43.31)	72,828 (41.97)	<0.0001
**Dyslipidemia (%)**	37,831 (21.94)	39,585 (22.07)	30,148 (21.91)	37,013 (21.33)	<0.0001
**Smoking (%)**					<0.0001
Non	62,433 (36.21)	60,707 (33.85)	45,611 (33.14)	55,081 (31.74)	
Ex	48,440 (28.1)	55,977 (31.21)	45,108 (32.78)	59,786 (34.45)	
Current	61,531 (35.69)	62,648 (34.93)	46,892 (34.08)	58,675 (33.81)	
**Drinking (%)**					<0.0001
Non	72,807 (42.23)	72,643 (40.51)	55,105 (40.04)	67,246 (38.75)	
Mild	78,318 (45.43)	84,052 (46.87)	65,070 (47.29)	83,509 (48.12)	
Heavy	21,279 (12.34)	22,637 (12.62)	17,436 (12.67)	22,787 (13.13)	
**Regular exercise (%)**	38,707 (22.45)	43,676 (24.35)	34,745 (25.25)	45,292 (26.1)	<0.0001
**Mean age, years**	59.66 ± 7.57	59.4 ± 7.61	59.78 ± 8.05	58.82 ± 7.61	<0.0001
**Weight, kg**	61.09 ± 7.94	65.58 ± 8	68.22 ± 8.31	72.66 ± 9.08	<0.0001
**BMI, kg/m²**	23.87 ± 2.9	23.95 ± 2.82	23.94 ± 2.82	24.02 ± 2.83	<0.0001
**Female**					
	**Q1** **(*n* = 119,** **651)**	**Q2** **(*n* = 115,** **897)**	**Q3** **(*n* = 123,** **090)**	**Q4** **(*n* = 117,** **377)**	***p* Value**
**Age, years (%)**					<0.0001
50–59	78,922 (65.96)	65,530 (56.54)	84,097 (68.32)	67,618 (57.61)	
60–69	28,187 (23.56)	36,502 (31.5)	28,114 (22.84)	35,889 (30.58)	
70–79	10,800 (9.03)	12,113 (10.45)	9012 (7.32)	11,982 (10.21)	
≥80	1742 (1.46)	1752 (1.51)	1867 (1.52)	1888 (1.61)	
**Income, low 25 (%)**	33,147 (27.7)	28,872 (24.91)	29,917 (24.3)	26,090 (22.23)	<0.0001
**DM (%)**	12,353 (10.32)	12,560 (10.84)	11,932 (9.69)	12,478 (10.63)	<0.0001
**Hypertension (%)**	44,005 (36.78)	43,059 (37.15)	41,543 (33.75)	41,234 (35.13)	<0.0001
**Dyslipidemia (%)**	36,010 (30.1)	35,231 (30.4)	35,614 (28.93)	34,012 (28.98)	<0.0001
**Smoking (%)**					<0.0001
Non	115,408 (96.45)	111,507 (96.21)	117,920 (95.8)	112,263 (95.64)	
Ex	1154 (0.96)	1240 (1.07)	1540 (1.25)	1622 (1.38)	
Current	3089 (2.58)	3150 (2.72)	3630 (2.95)	3492 (2.98)	
**Drinking (%)**					<0.0001
Non	102,496 (85.66)	99,507 (85.86)	103,995 (84.49)	100,400 (85.54)	
Mild	16,478 (13.77)	15,729 (13.57)	18,299 (14.87)	16,250 (13.84)	
Heavy	677 (0.57)	661 (0.57)	796 (0.65)	727 (0.62)	
**Regular exercise (%)**	19,863 (16.6)	21,781 (18.79)	24,659 (20.03)	24,830 (21.15)	<0.0001
**Mean age, years**	58.56 ± 7.72	59.23 ± 7.79	57.52 ± 7.35	58.57 ± 7.74	<0.0001
**Weight, kg**	52.5 ± 7.18	55.42 ± 7.18	57.7 ± 7.28	60.66 ± 7.82	<0.0001
**BMI, kg/m²**	23.98 ± 3.09	23.8 ± 3	23.61 ± 2.92	23.43 ± 2.91	<0.0001

Data are presented as mean ± standard deviation or proportion (%). Q, quartile; BMI, body mass index; DM, diabetes mellitus.

**Table 3 jcm-13-00092-t003:** Risk of knee osteoarthritis according to height quartile.

Height	Number	Knee OA	IR/1000 (Person Years)	Hazard Ratio (95% CI)				
				Model 1	Model 2	Model 3	Model 4	Model 5
**Total**								
**Q1**	292,055	121,065	54.8360	1 (ref.)	1 (ref.)	1 (ref.)	1 (ref.)	1 (ref.)
**Q2**	295,229	120,743	53.5730	0.977 (0.969, 0.985)	0.985 (0.977, 0.993)	0.985 (0.978, 0.993)	0.99 (0.982, 0.998)	0.908 (0.9, 0.915)
**Q3**	260,701	109,290	55.0482	1.004 (0.996, 1.012)	0.975 (0.967, 0.983)	0.976 (0.968, 0.984)	0.988 (0.979, 0.996)	0.853 (0.846, 0.86)
**Q4**	290,919	118,414	52.8754	0.964 (0.956, 0.972)	0.979 (0.971, 0.987)	0.98 (0.973, 0.988)	0.996 (0.988, 1.004)	0.787 (0.781, 0.795)
**Gender**								
**Male**								
**Q1**	172,404	56,517	40.9737	1 (ref.)	1 (ref.)	1 (ref.)	1 (ref.)	1 (ref.)
**Q2**	179,332	57,798	39.7050	0.968 (0.957, 0.979)	0.974 (0.963, 0.985)	0.973 (0.962, 0.985)	0.971 (0.96, 0.983)	0.904 (0.893, 0.915)
**Q3**	137,611	44,354	39.8483	0.972 (0.96, 0.984)	0.966 (0.954, 0.978)	0.965 (0.953, 0.977)	0.963 (0.951, 0.975)	0.855 (0.844, 0.866)
**Q4**	173,542	55,229	38.8388	0.947 (0.936, 0.958)	0.968 (0.957, 0.979)	0.967 (0.955, 0.978)	0.964 (0.952, 0.975)	0.796 (0.786, 0.807)
**Female**								
**Q1**	119,651	64,548	77.9174	1 (ref.)	1 (ref.)	1 (ref.)	1 (ref.)	1 (ref.)
**Q2**	115,897	62,945	78.8670	1.012 (1.001, 1.023)	0.998 (0.987, 1.009)	0.999 (0.989, 1.011)	1.013 (1.002, 1.024)	0.917 (0.907, 0.927)
**Q3**	123,090	64,936	74.4442	0.956 (0.946, 0.966)	0.977 (0.966, 0.988)	0.98 (0.969, 0.99)	1.005 (0.995, 1.017)	0.848 (0.839, 0.858)
**Q4**	117,377	63,185	77.2917	0.992 (0.981, 1.003)	0.992 (0.981, 1.002)	0.996 (0.985, 1.007)	1.035 (1.023, 1.046)	0.787 (0.778, 0.796)

Data are presented as number; OA, osteoarthritis; IR, incidence rate; CI, confidence interval; Q, quartile; Model 1: unadjusted; Model 2: adjusted for age, sex; Model 3: adjusted for age, sex, income, smoking, drinking, exercise, diabetes mellitus, hypertension, dyslipidemia; Model 4: model 3 plus body mass index; Model 5: model 3 plus weight.

**Table 4 jcm-13-00092-t004:** Risk of knee osteoarthritis according to height quartile stratified by age.

Total					Hazard Ratio (95% CI)		
Age	Height	Number	Knee OA	IR/1000 (Person-Years)	Model 3 (*p* < 0.0001)	Model 4 (*p* < 0.0001)	Model 5 (*p* < 0.0001)
50–59	Q1	177,671	69,410	48.5214	1 (ref.)	1 (ref.)	1 (ref.)
	Q2	167,689	61,536	44.7085	0.967 (0.956, 0.977)	0.974 (0.963, 0.984)	0.894 (0.884, 0.904)
	Q3	159,388	62,514	48.5412	0.963 (0.953, 0.974)	0.979 (0.968, 0.99)	0.848 (0.838, 0.857)
	Q4	166,985	60,853	44.2354	0.965 (0.954, 0.975)	0.983 (0.972, 0.994)	0.778 (0.769, 0.787)
60–69	Q1	81,248	37,765	65.4846	1 (ref.)	1 (ref.)	1 (ref.)
	Q2	91,719	43,477	66.3163	0.987 (0.974, 1.001)	0.99 (0.976, 1.003)	0.905 (0.893, 0.918)
	Q3	68,686	32,848	66.7852	0.989 (0.975, 1.004)	0.995 (0.981, 1.01)	0.859 (0.847, 0.872)
	Q4	90,196	42,413	64.9676	0.978 (0.964, 0.992)	0.991 (0.977, 1.005)	0.785 (0.774, 0.797)
70–79	Q1	29,135	12,751	70.3615	1 (ref.)	1 (ref.)	1 (ref.)
	Q2	31,710	14,429	71.9773	1.036 (1.011, 1.061)	1.036 (1.012, 1.061)	0.949 (0.927, 0.972)
	Q3	28,502	12,604	68.4241	1.039 (1.014, 1.065)	1.043 (1.017, 1.069)	0.898 (0.876, 0.921)
	Q4	29,517	13,711	72.7004	1.044 (1.019, 1.07)	1.049 (1.024, 1.074)	0.83 (0.81, 0.851)
≥80	Q1	4001	1139	58.8918	1 (ref.)	1 (ref.)	1 (ref.)
	Q2	4111	1301	60.9286	1.067 (0.986, 1.156)	1.065 (0.984, 1.153)	0.976 (0.901, 1.057)
	Q3	4125	1324	61.7391	1.066 (0.985, 1.154)	1.064 (0.983, 1.151)	0.919 (0.849, 0.995)
	Q4	4221	1437	64.1502	1.127 (1.043, 1.218)	1.125 (1.041, 1.216)	0.891 (0.824, 0.963)

Data are presented as numbers; OA, osteoarthritis; IR, incidence rate; CI, confidence interval; Q, quartile; Model 3: adjusted for age, sex, income, smoking, drinking, exercise, diabetes mellitus, hypertension, dyslipidemia; Model 4: model 3 plus body mass index; Model 5: model 3 plus weight.

## Data Availability

The datasets used in this study are contained within the main article.

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
