# Peer review of "Decreased Risk of Knee Osteoarthritis with Taller Height in an East Asian Population: A Nationwide Cohort Study"

_jcm, 2023, doi:10.3390/jcm13010092_

Round 1

Reviewer 1 Report

Comments and Suggestions for Authors

Very Respected authors,

After carefully reading your manuscript, I can say that the Abstract is well written as the Introduction of the manuscript. The objective is clear and the section Material and Methods are written in detail. The findings are well presented in the section Results. The section Discussion can improve. Please, write more about your study and your findings in the first paragraph of this section. Two or three sentences in the first paragraph generally are enough. Than write about your research. Please, check the second reference in the literature: Kan at al...

Author Response

Thank you for your kind advice. Following your guidance, we have added content to the first paragraph of the discussion, which we believe has clarified the message. Additionally, we have included the information for the second reference. However, during the process of changing to the EndNote format, it seems that some information might not be visible. We will review and confirm this later. Thanks again.

The attachment contains the same content.

Reviewer 2 Report

Comments and Suggestions for Authors

Thank you to the authors for an interesting and very well-prepared article. 

The introduction introduces the reader well to the research topic, the research methods are clearly described, the results are presented in a clear way, the discussion is correct.

Two minor technical notes:

1. I would attach a table with supplementary material to the article,

2. figure 1 is of poor quality (resolution).

Author Response

  1. I would attach a table with supplementary material to the article,

Answer]

 Thank you for your guidance. As you suggested, we have incorporated the content from the supplementary material into the main text, now presented as Table 1.

  1. figure 1 is of poor quality (resolution).

Answer]

 Thank you for your advice. We have replaced Figure 1 with a higher resolution version. Your guidance is much appreciated.

The attachment contains the same content.

Reviewer 3 Report

Comments and Suggestions for Authors

The aim of the authors in this study was to investigate the relationship between height and knee osteoarthritis in Korean adults. The noteworthy point is Big Data, but there are serious uncertainties.

Major problems:

1.      The prominent point in this study is the use of big data, but the data does not include effective and important information on knee osteoarthritis. In arthritis, any factor affecting knee loading should not be ignored. History of knee loading during daily activities, job status, alignment and posture of lower limb joints, and trauma and diseases are very important, which are not included in this study. In addition, the condition of knee arthritis is not clear at all, it is not possible to calculate the effect of correlation and risk factors, but the severity of the injury and the grade of arthritis are not taken into account. Ignoring these key parameters may lead to unreliable results and incorrect conclusions.

2.      Another point is that the difference between the lowest and highest height in both groups of women and men is less than 10 cm, and this shows that the height difference in the four quartiles is not considerable. This point should be noted both in the results and in the discussion and may be the reason for the lack of significance in most of the used models.

Minor problems:

1.      Considering that the authors have investigated the relationship between height and the prevalence of knee arthritis, it should be mentioned about the knee joint in the title, hypothesis, and statement of the problem.

2.      It should be mentioned of HR in the abstract.

3.      The study of Finland cannot be mentioned in the abstract, while the reference cannot be mentioned.

4.      The novelty of the study is not clear. The existence of a contradiction is not innovation because the result of this study will also tend to one side.

5.      ​According to the purpose of the study, it is better to refer to related studies in the introduction instead of stating the relationship between some factors such as diabetes and arthritis, and mention the target population and race, and what is the effect of height on knee osteoarthritis? What exactly is study innovation? Is the importance of this study only big data?

The ethics registration code is not reported.

a.      The definition of the quartiles is not in the text of the manuscript, and it is in the supplementary file, which is recommended to be added to the manuscript.

b.      ​In lines 129 to 132: it is stated "Although there were differences in the total number of participants among the quartiles, most lifestyle indicators, comorbidities, smoking, and percentage of drinking were similar" However in Tables 1 and 2 between quartiles in terms of the above parameters there are the significant difference. Explain about this.

c.      ​In the caption of Figure 2, A, B, and C should be defined. The graphs show that with increasing age, the prevalence of knee arthritis increases in all cases, but there are slight differences between the four height quartiles, which of course have not been determined that are significant or not.
